# The Response to DNA Damage at Telomeric Repeats and Its Consequences for Telomere Function

**DOI:** 10.3390/genes10040318

**Published:** 2019-04-24

**Authors:** Ylli Doksani

**Affiliations:** IFOM, The FIRC Institute of Molecular Oncology, via Adamello 16, 20139 Milan, Italy; ylli.doksani@ifom.eu; Tel.: +39-02-574303258

**Keywords:** telomere maintenance, shelterin complex, end-protection problem, telomere damage, telomeric double strand breaks, telomere replication, alternative lengthening of telomeres

## Abstract

Telomeric repeats, coated by the shelterin complex, prevent inappropriate activation of the DNA damage response at the ends of linear chromosomes. Shelterin has evolved distinct solutions to protect telomeres from different aspects of the DNA damage response. These solutions include formation of t-loops, which can sequester the chromosome terminus from DNA-end sensors and inhibition of key steps in the DNA damage response. While blocking the DNA damage response at chromosome ends, telomeres make wide use of many of its players to deal with exogenous damage and replication stress. This review focuses on the interplay between the end-protection functions and the response to DNA damage occurring inside the telomeric repeats, as well as on the consequences that telomere damage has on telomere structure and function.

## 1. Telomere Structure and End-Protection Functions

Mammalian telomeres are made of tandem TTAGGG repeats that extend several kilobases and terminate with a 3′ single-stranded overhang about 50–400 nucleotides long. Telomeric repeats are coated in a sequence-specific fashion by a 6-protein complex named shelterin that prevents the activation of the Double Strand Break (DSB) response at chromosome ends [1,2,3]. Two shelterin components, TRF1 and TRF2, bind double-stranded telomeric repeats through their MYB domains and recruit the rest of the complex at telomeres by interacting with TIN2, which interacts with TPP1, which interacts with POT1 (Figure 1A). The POT1 gene also binds directly to single-stranded TTAGGG repeats through its OB-fold domains [4]. The POT1 gene has undergone duplication in rodents which express two POT1 proteins, POT1a and POT1b [5,6]. A sixth protein, RAP1, is recruited to telomeres via direct interaction with TRF2. 

Shelterin deploys different components to deal with distinct aspects of the DSB response at telomeres. The component TRF2 blocks two of the earliest steps of the DSB response, ATM signaling and Non-Homologous End Joining (NHEJ), which are activated immediately after the exposure of DNA ends. Removal of TRF2 from telomeres results in ATM activation and massive chromosome fusions, without loss of telomeric repeats [8,9,10,11,12]. The same phenotype is not observed after deletion of other shelterin components. Telomeres deprived of POT1 activate the ATR pathway [5,12,13]. ATR responds to RPA-coated single-stranded DNA that normally accumulates after resection of DSBs in S/G2 phases or during replication stress [14,15,16]. POT1 has been involved in facilitating telomere replication; therefore, ATR activation in POT1-deleted cells could be in part attributable to telomere replication stress [17,18]. However, ATR is also activated at telomeres when POT1 is depleted in G1 cells, indicating that POT1 prevents the exposure of existing single-stranded DNA at telomeres (i.e., the 3′ overhang) [19]. As POT1 binds single-stranded TTAGGG repeats, its tethering at the 3′ overhang by the shelterin complex would effectively prevent RPA loading and thus ATR activation at telomeres [20,21].

The mechanism by which TRF2 blocks ATM signaling and NHEJ is more intricate. Purified TRF2 has the unique ability to remodel telomeric DNA into a t-loop structure, formed by the invasion of the 3′-overhang into the internal repeats (Figure 1B) [22,23,24]. T-loop structures form at telomeres of different organisms and are lost upon TRF2 deletion from mouse cells [22,25,26,27,28,29,30,31]. T-loop formation sequesters the chromosome end, preventing its association with the DNA-end sensors (Mre11, Rad50, Nbs1), MRN complex and (Ku70/80) Ku complex and therefore preventing activation of the ATM signaling and NHEJ at chromosome termini. In addition to t-loop formation, TRF2 curbs the DSB response by blocking the accumulation of the RNF168 ubiquitin ligase at telomeres via a small iDDR (inhibitor of the DNA Damage Response) region, located in the hinge domain of TRF2 [32]. Furthermore, it has been suggested that TRF2 can prevent the Ku70/80 complex from initiating end-joining, by directly binding and blocking a surface of Ku70 that mediates tetramerization of Ku heterodimers, required for end-joining [33].

## 2. End-Protection vs. the Response to DNA Damage Inside the Telomeric Repeats

Shelterin has the ability to bind throughout the telomeric repeats as a complex, and it is abundant enough to cover all telomeres [34]. If a shelterin component works by directly blocking a DNA damage response factor (e.g., via an inhibitory interaction), such inhibitory effect would spread throughout the telomeric repeats. The iDDR region of TRF2 and its inhibition of Ku tetramerization, described above, are predicted to act in this fashion. Also the POT1 tethering mechanism predicts repression of ATR signaling throughout the telomeric repeats, but only if the top (G-rich) strand is exposed as single-stranded. On the other hand, the t-loop model predicts inhibition of the DSB response only at the chromosome terminus (I will refer to this as an end-specific mechanism). Other end-specific mechanisms, involving a secondary structure or an end-binding protein, have also been described [35,36,37].

How, then, does shelterin affect the response to DNA damage occurring inside the telomeric repeats?

## 3. ATM Activation at Damaged Telomeres

Although TRF2 is required to repress ATM signaling at chromosome ends, its presence is not sufficient to fully inhibit ATM throughout the telomeric repeats. Common markers of the DNA damage response, 53BP1 foci have been detected at telomeres after Ionizing Radiation (IR), treatment with DNA damaging agents and also after DSBs were induced in the telomeric repeats by the FokI nuclease tethered to TRF1 (FokI-TRF1) or by Cas9 [38,39,40,41]. At least in mouse cells, accumulation of 53BP1 at telomeric DSBs was shown to be dependent on the ATM kinase and was not due to a loss of TRF2 or of telomeric repeats [40]. Furthermore, in the absence of the nuclease Apollo, some telomeres activate ATM signaling, despite the presence of TRF2 (likely due to a failure to generate the 3′ overhang on the chromatid replicated by the leading strand) [42,43,44]. Therefore, full repression of ATM signaling at chromosome ends seems to be achieved through a TRF2-dependent, end-specific mechanism, like the formation of t-loops (Figure 2).

## 4. C-NHEJ Repression at Telomeres

Although t-loop formation would also block c-NHEJ at chromosome ends, several studies have reported additional mechanisms by which c-NHEJ is inhibited at telomeres [31,32,33,38,45,46,47,48]. These mechanisms are predicted to repress c-NHEJ also at DSBs occurring inside the telomeric repeats. A tight repression of c-NHEJ at telomeres is consistent with experiments showing that the c-NHEJ pathway does not contribute to the repair of telomeric DSBs, generated by FokI-TRF1, although those breaks accumulated preferentially in S-phase, when c-NHEJ competes with homology-based repair [40]. Repression of c-NHEJ throughout the telomeric repeats is unlikely to be achieved by de novo formation of a t-loop on the proximal end of a telomeric DSB. This is because c-NHEJ normally precedes 5′-end resection, which is required to generate the 3′-overhang necessary for t-loop formation. Shelterin therefore seems to repress c-NHEJ in a redundant fashion, by an end-specific mechanism (i.e., the t-loop), as well as by general inhibition throughout the telomeric repeats (e.g., via its iDDR region and possibly through an interaction with Ku70) (Figure 2). Although both the t-loop and the general repression mechanisms require TRF2, the latter seems to involve also RAP1, which is recruited at telomeres by TRF2 [47,49]. This redundancy in blocking c-NHEJ at telomeres seems justified by the irreversible nature of chromosome fusions and the severe consequences they can have for genome integrity [50].

## 5. Alt-NHEJ for the Repair of Telomeric DSBs

A strict blockade on c-NHEJ does not necessarily preclude DSB repair inside the telomeric repeats. Because of the unique sequence context, minimal resection of telomeric DSBs will always generate perfect homology of the exposed 3′ single-stranded tails (Figure 3), making them an ideal substrate for alternative-NHEJ (alt-NHEJ). The alt-NHEJ pathway relies on the poly (ADP-ribose) polymerase 1 (PARP1) and microhomology to repair DSBs using either Ligase 1 or 3 [51,52]. Consistent with this view, repair of DSBs inside the telomeric repeats induced by the FokI-TRF1 fusion protein, is reduced in the presence of the PARP inhibitor Olaparib, in PARP1-deleted cells, or after knockdown of Ligase 3 [40]. In this view, telomeric DSBs behave differently from genome-wide DSBs, where a contribution of the alt-NHEJ pathway emerges only in c-NHEJ-defective backgrounds [53,54]. These results suggest that alt-NHEJ could be favored inside telomeres, due to their highly repetitive nature. There is a great advantage in adopting a microhomology-directed end-joining at telomeric repeats: directionality in the repair of broken telomere ends. Annealing of complementary overhangs will favor correct joining of broken ends in a head-to-tail orientation, over incorrect head-to-head telomere joining that generates chromosome fusions. If re-ligation of broken ends is delayed, full engagement of the resection machinery at telomeric DSBs will generate longer, complementary overhangs that are the perfect substrate for Rad52-dependent Single-Strand Annealing repair (SSA) [55,56]. A role for the SSA pathway in the repair of telomeric DSBs is consistent with recent reports suggesting an important function for Rad52 in telomere maintenance in Alternative Lengthening of Telomeres (ALT) cells that does not involve generation of classical intermediates of homologous recombination [57,58].

Apart from contributing to the repair of DSBs inside the telomeric repeats, alt-NHEJ can also generate chromosome fusions in cells with dysfunctional telomeres [59,60]. Unlike c-NHEJ, however, the alt-NHEJ pathway is fully unleashed at chromosome ends upon complete removal of shelterin from telomeres (by co-deletion of TRF1 and TRF2) in Ku-deficient cells. Therefore, alt-NHEJ seems to be repressed only at chromosome termini, in a redundant fashion by different shelterin components as well as by the Ku complex, which is a general repressor of this pathway genome-wide [60,61]. TRF2 could inhibit alt-NHEJ at the chromosome terminus by t-loop formation, whereas in the absence of TRF2, the recently discovered role of TIN2 in limiting accumulation of PARP1 at telomeres could contribute to the protection from alt-NHEJ at chromosome termini [62].

Given its contribution to chromosome fusions, why is alt-NHEJ not fully repressed throughout the telomeric repeats in the same way that c-NHEJ is? One explanation could lie in the different kinetics of the two processes [60,63]. It is possible that in the absence of perfect homology to guide the repair (as happens at DSBs inside telomeric repeats), alt-NHEJ is too slow to represent a significant threat to functional telomeres, where the half-life of the unprotected form (e.g., when t-loops are resolved during replication) is short. Importantly, alt-NHEJ seems to contribute substantially to the fusion of critically short telomeres in telomerase-negative cells [59,64,65,66]. It is possible that, as telomeres become critically short, the mechanisms that repress alt-NHEJ at chromosome ends are lost before those that repress c-NHEJ.

## 6. Homologous Recombination during Telomere Replication

Homologous Recombination (HR) is primarily used to deal with replication stress and DSBs in the S/G2 phases of the cell cycle. In these conditions, there is a strong preference towards using the homologous sequence on the sister chromatid as a template to repair a DSB or bypass lesions during DNA replication [67]. Typically, this process results in the formation of double Holliday Junctions (HJ) between sister chromatids, that can either be “dissolved” by the concerted action of Blm/TopIII/Rmi1 or resolved through the action of HJ resolvases such as Mus81 or Slx1/Slx4 [68]. Resolution by HJ resolvases will often result in Sister Chromatid Exchanges (SCEs), that can be detected in metaphase spreads by differential staining, or Chromosome Orientation FISH (CO-FISH) [69]. The rate of spontaneous SCEs genome-wide, estimated across different organisms, ranges from 2–7 events/genome [70,71,72,73,74]. On the other hand, around 1–4% of mouse or human telomeres experience spontaneous SCEs (named T-SCEs) [75,76,77,78,79,80]. Based on these numbers and considering the size of the telomeric repeats, the incidence of T-SCEs is several-fold higher than the incidence of SCEs genome-wide. A few studies have tackled this question directly and, accommodating for experimental variations, have estimated a 20- to over 1600-fold higher incidence of SCEs at telomeres compared to the rest of the genome [81,82,83]. Therefore, sister chromatid recombination is frequently used at telomeric repeats, probably to bypass obstacles during telomere replication. It seems reasonable to imagine that, in the presence of functional sister chromatid cohesion, the benefits of using sister chromatid recombination during telomere replication outweigh the risk of having limited unequal exchanges between telomeric repeats.

It is important to note that sister chromatid recombination could also initiate at gaps or nicks, without the generation of a DSB intermediate [84,85,86,87]. Despite this caveat, genetic analysis of the pathways that repress HR at telomeres has mainly relied on the frequency of T-SCEs as a readout of telomere recombination (also for lack of better assays). T-SCEs accumulate in Ku-deficient backgrounds at telomeres lacking either RAP1 or POT1a and POT1b in mouse cells [77,78]. Therefore, the functions of both RAP1 and POT1 are required to limit T-SCEs, in a redundant fashion with the Ku complex. It is not clear whether this increase in T-SCEs also reflects a role of these proteins in inhibiting recombination initiation at chromosome ends (with the telomeric 3′-overhang invading other telomeres). Unlike sister-chromatid recombination, these telomere-end-initiated events would significantly affect telomere length by engaging in break-induced replication or unequal exchanges of telomeres from different chromosomes [7]. In normal cells, however, chromosome termini are not recombinogenic and telomere length is relatively stable, with no significant recombination between telomeres of different chromosomes [88]. How this protection is achieved is not clear, but it seems to occur only through an end-specific mechanism. Indeed, generation of DSBs at telomeric repeats is sufficient to induce T-SCEs in cells with functional shelterin [40,41]. In principle, t-loop formation would prevent strand invasion at other telomeres, although the t-loop structure itself needs to be protected from engaging in intramolecular recombination that could lead to telomere length variation [7,89]. The POT1 binding to the 3′-telomeric overhang could also repress recombination, by preventing the formation of Rad51-coated nucleofilament that initiates strand invasion. Further studies that focus on recombination events initiated at the chromosome termini, could help clarify this important aspect of end-protection.

## 7. Telomere Loss, Beyond the End-Replication Problem

Telomere erosion is highly relevant to age-associated pathologies and tumorigenesis. Critically short telomeres activate the DNA damage response, which can lead to cellular senescence or apoptosis. This proliferative barrier induced by telomere shortening contributes to tumor suppression, but also to ageing phenotypes by limiting tissue regeneration [90,91,92,93]. Telomere shortening also plays an opposite role in tumorigenesis, when cell-cycle checkpoints are lost and cells with critically short telomeres proliferate. During this period, known as telomere crisis, short telomeres fuel genome instability by initiating a wide spectrum of cancer-relevant genome alterations (reviewed in reference [94]).

Telomere shortening is due to the inability of the DNA replication machinery to copy the last few nucleotides of the lagging strand (the end-replication problem); however, telomere shortening is substantially affected by 5′-end resection and telomere damage [95,96,97]. Telomerase activity counteracts erosion and maintains telomere length in the germline. However, in somatic cells, where telomerase activity is reduced or absent, the only mechanisms that counteract telomere loss are those that assist telomere replication and repair telomere damage. Several studies indicate that telomere damage can induce telomere shortening either directly or by interfering with telomere replication [98,99,100,101,102].

Telomeres are replicated by terminal forks (i.e., replication forks moving towards a DNA end), and if they collapse, replication cannot be rescued by forks arriving from nearby origins, as happens in internal regions. Therefore, fork collapses within the telomeric repeats can result in telomere truncation or whole telomere loss (Figure 4A). This scenario is aggravated by the fact that telomeres are notoriously difficult to replicate and behave like the common fragile sites where replication problems occur [103,104]. Telomeric repeats present several challenges to the replisome, like the propensity to form secondary structures (G-quartets), ongoing transcription (TERRA) or the t-loop structure (Figure 4B). Indeed, single molecule analysis has shown frequent fork stalling at telomeric repeats, and shelterin plays an essential and highly conserved role in promoting telomere replication. Deletion of Taz1, the *Schizosaccharomyces pombe* orthologue of TRF1 and TRF2, leads to frequent fork stalling at telomeric repeats and massive telomere loss [105]. In mammalian cells, TRF1 deletion severely impacts telomere replication, resulting in frequent fork stalling, ATR activation and accumulation of abnormalities in metaphase spreads, like fragile telomeres and sister telomere association [103,104,106]. Telomere replication constitutively requires specialized factors that assist replication fork progression. Deficiencies in RTEL1 or in the RecQ helicases WRN and BLM are associated with increased telomere fragility and telomere loss [104,107,108,109]. These helicases could promote replication fork progression at telomeres by resolving secondary structures like the G4-DNA and the t-loop, but other relevant activities cannot be excluded. Despite their inherent ability to respond to replication stress genome-wide, the action of these factors at telomeres is, at least in part, dependent on their direct association with shelterin. TRF1, for example, interacts with the BLM helicase, and this binding is important in preventing fragility of the telomeres replicated by the lagging-strand polymerase [108]. TRF2 interacts with RTEL1 in S-phase, and this interaction is required to prevent telomere loss [110]. TRF2 has also been shown to interact with the WRN helicase, supporting an important contribution of TRF2 to telomere replication [111,112]. Association of these helicases with shelterin might be important in locally increasing their concentration at telomeres, but it could also have a qualitative impact by directing/modulating their activity towards specific structures that need to be resolved. Apart from the above-mentioned proteins, shelterin interacts with a long list of factors involved in DNA replication and DNA damage response, although the contribution of many of these interactors in telomere maintenance is not clear (for a review see references [3,113]). It is possible that, in some cases, mutations in shelterin interactors that increase the rate of stochastic telomere loss may be dismissed because of a lack of effect on the overall telomere population. Average telomere length, however, cannot be an absolute criterion, because there is clear evidence that one or a few critically short telomeres are sufficient to induce both senescence and genome instability [114,115,116]. Identification of new pathways that prevent telomere loss could benefit from the use of specific assays that can detect single telomere truncation events in a population [117,118].

## 8. Telomere Elongation by Homologous Recombination

Although DNA damage response pathways can counteract stochastic telomere loss, ultimately long-term telomere maintenance is ensured by telomerase. In some cases, however, telomerase-negative cells acquire the ability to maintain telomeres using homologous recombination pathways. Alternative lengthening of telomeres was first observed in yeast cells that survived telomerase deletion and then found as a telomere maintenance mechanism in a substantial number of human cancers [119,120,121]. The ALT cell lines have very heterogenous telomeres, high rates of T-SCEs and frequent exchanges between non-sister telomeres [88,122]. Telomere maintenance in these cells is impaired after knockdown of genes like Brca2, FACD2, FANCA, SMC5/6 BLM and Rad52, consistent with HR-based telomere elongation [57,58,123] reviewed in reference [124]. How ALT cells evolve is not entirely clear, but the process results in frequent use of HR at telomeres, rather than a general upregulation of HR throughout the genome [125]. Increased telomere recombination does not seem to derive from a loss of function in shelterin as neither mutations in shelterin components nor changes in their expression levels were found in ALT cells [126]. The most common mutation found in ALT cells is a loss of function in the ATRX/DAXX chromatin remodeling complex, involved in depositing the histone H3.3 to telomeres [126,127]. The link between ATRX mutation and ALT is still under investigation. Loss of ATRX alone is not sufficient to induce ALT, whereas restoration of ATRX suppressed some ALT phenotypes [128].

The ALT telomeres contain frequent nicks and gaps that could initiate telomere recombination via induction of replication stress and frequent DSB formation [123,129]. Supporting this view, telomere damage can trigger ALT features in telomerase-positive cells, and induction of telomeric DSBs can initiate inter-chromosomal telomere recombination in ALT cells [40,41,130,131,132]. The development of the ALT pathway may therefore involve the selection of cells that experience telomere damage, although the initial source of telomere damage in ALT cells is not clear. One possibility is that telomeric DNA damage arises from common metabolic intermediates, like reactive oxygen species, that preferentially target G-rich sequences [97,133]. Initial telomere damage could induce replication stress at telomeres, which in turn leads to accumulation of secondary damage that fuels ALT activity.

One of the most consistent features of ALT cells is the presence of Extra-Chromosomal Telomeric Circles (ECTC) that are either double-stranded (commonly referred to as t-circles) or partially single-stranded (commonly referred to as C-circles) [89,134,135]. Despite its common use, t-circle stands for telomeric circles; C-circles are telomeric circles with an intact C-strand and with one or more single-stranded gaps on the G-strand, whereas G-circles (less abundant) are telomeric circles with an intact G-strand and with one or more single-stranded gaps on the C-strand [135,136]. T-circles can be detected in two-dimensional agarose gels or through an in vitro rolling-circle replication assay and are not a unique feature of ALT cells. They accumulate in a TRF2 mutant lacking the N-terminal basic domain, which is involved in binding and protecting Holliday Junctions [62,89,137,138]. Given the role of TRF2 in formation/maintenance of t-loops, t-circles in the TRF2 delta basic mutant could form as a result of t-loop excision, in a reaction that resembles the resolution of a Holliday Junction [89]. T-circles have also been detected in normal cells with long telomeres and in a long list of mutants, apparently unrelated to t-loop metabolism, suggesting alternative mechanisms of t-circle formation [139,140,141,142,143,144,145]. Given that telomeric damage and replication stress can induce both t- and C-circles in non-ALT cells, it is possible that ECTC are a direct consequence of the accumulation of telomere damage during the establishment of ALT [142,146,147,148,149].

The use of post-replicative HR-based sequence exchanges between telomeres would generate both long and short telomeres without a net gain in telomeric repeats. The ALT cells seem to adopt specialized recombination pathways that involve extensive synthesis of telomeric repeats. This process occurs spontaneously in ALT cells and is greatly stimulated by the induction of telomeric DSBs [131]. Furthermore, DSB-induced telomeric DNA synthesis occurred independently of Rad51 and relied on the replisome components PCNA, RFC and DNA polymerase delta. This pathway resembled the Break-Induced Replication (BIR) pathway of telomere maintenance, described first in yeast (reviewed in reference [150]). Break-induced replication can initiate after strand invasion of a telomere end to another telomere and initiate conservative telomere synthesis through the bubble migration mechanism (Figure 5) [151,152].

How strand invasion occurs in the absence of Rad51 is not clear, although the abundance of homology and the presence of gaps at ALT telomeres might favor strand annealing in the absence of Rad51. Break-induced telomere synthesis in ALT cells seems to proceed for very long distances, with many events reaching over 70 kb [131]. It is possible that at least part of these super-elongation events occur through a rolling circle mechanism, where an ECTC serves as a template for telomere elongation (Figure 5) [150]. Structures compatible with rolling circle intermediates of telomere elongation have been observed in the yeast *Candida parapsilosis* and a rolling circle-like mechanism could account for in-cis telomere elongation events observed in ALT cells [153,154]. Development of techniques that monitor the intermediates of telomere elongation in ALT cells could provide important insights on the prevalence of the different mechanisms of telomere elongation in ALT cells.

## 9. Concluding Remarks

The evolution of end-protection mechanisms has left ample room for the DNA damage response to act within the telomeric repeats. Rather than being a general repressor, shelterin seems to act more like a puppeteer of the DDR show at telomeres. Decades of studies in the telomere field have shown that both telomere function and genome stability can be severely impacted by telomere damage and replication stress. It is now becoming clear that accumulation of telomere damage plays an active role in telomere maintenance in ALT cancer cells. Yet, the molecular functions of numerous DDR factors that are actively recruited at telomeres are not known. The development of new strategies that allow a more careful analysis of telomere length and structure, combined with the genetic tools offered by the CRISPR technology, will help to understand better the mechanisms of the DNA damage response at telomeric repeats and their contribution to the maintenance of genome stability.

## Figures and Tables

**Figure 1 genes-10-00318-f001:**
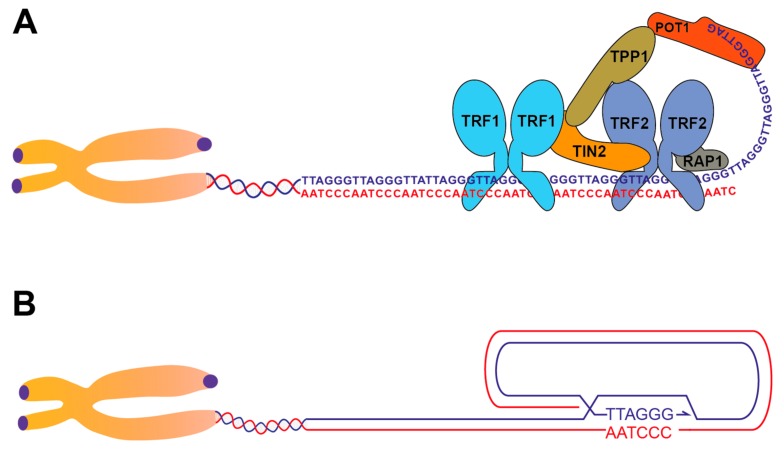
Telomere structure and the shelterin complex. (**A**) Illustration of the shelterin complex loaded on the telomeric DNA. Shelterin components TRF1 and TRF2 are shown as dimers. For the sake of simplicity only one complex is shown, although many complexes bind throughout the telomeric repeats. (**B**) Illustration of the telomere in the t-loop configuration, where the 3′ telomeric overhang has invaded the internal repeats, pairing with the complementary C-rich strand. The actual structure at the base of the t-loop is not known and multiple configurations are possible [7].

**Figure 2 genes-10-00318-f002:**
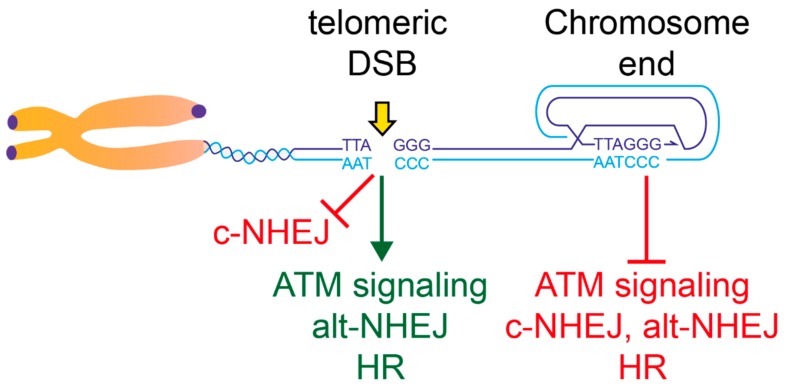
End-protection and the response to telomeric Double Strand Breaks (DSBs). This model summarizes the data on the control of the different aspects of the DSB response throughout the telomeric repeats. Active pathways are shown in green, whereas repressed ones are shown in red. The chromosome terminus is shown in a t-loop configuration. Although the t-loop structure could provide protection from all the pathways indicated, redundant repression mechanisms exist for c- and alt-Non-Homologous End Joining (NHEJ) pathways at chromosome ends. See text for details.

**Figure 3 genes-10-00318-f003:**
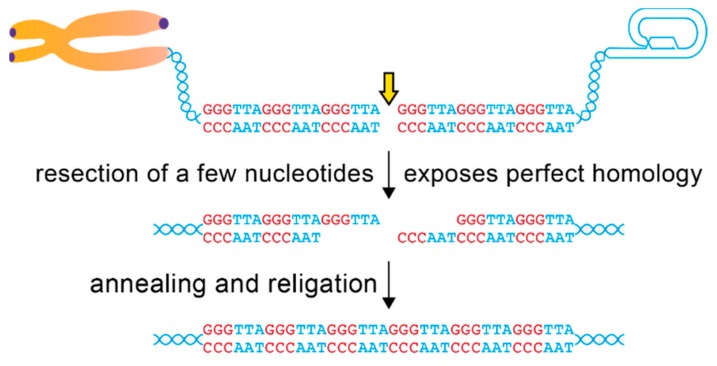
Repair of telomeric DSBs through alt-NHEJ. Illustration of alt-NHEJ repair of a telomeric DSB. Processing of telomeric DSBs generates perfectly cohesive ends. Telomeric repeats are shown in two colors for visual purposes. Note that even if the two broken ends become dissociated, homology-guided repair will always promote correct rejoining of a broken telomere piece over generation of telomere fusions.

**Figure 4 genes-10-00318-f004:**
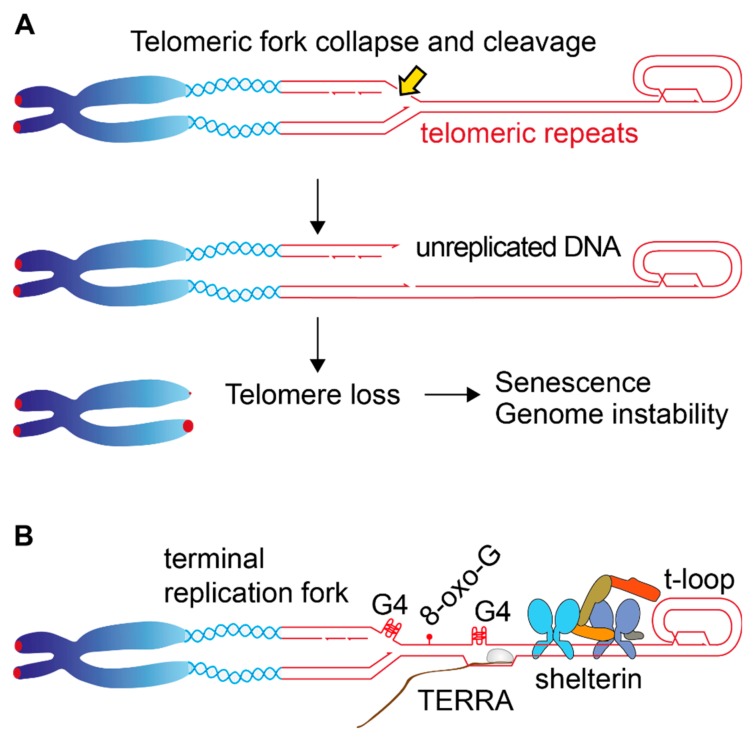
Telomere loss due to replication problems. (**A**) Illustration of the terminal fork entering the telomeric repeats. Possible structural transitions like fork reversion and processing are not shown. In the event of nucleolytic cleavage of the telomeric fork, the distal part of the sister telomere will not be replicated, resulting in abrupt telomere truncation. (**B**) Potential obstacles that the replication fork can encounter on the telomeric repeats. The G-rich telomeric repeats have a tendency to form G-quartets (G4) that can interfere with replication. The abundance of Guanines makes telomeres more susceptible to oxidative stress (8-oxo-G) compared to other genomic regions. Telomeres are continuously transcribed by PolII to generate the Telomeric-Repeat-Containing RNA (TERRA), a process that can potentially interfere with replication fork progression. Other potential obstacles to the replication fork are the presence of the tightly-bound shelterin complex (for simplicity, only one complex is shown in the figure) and the t-loop structure.

**Figure 5 genes-10-00318-f005:**
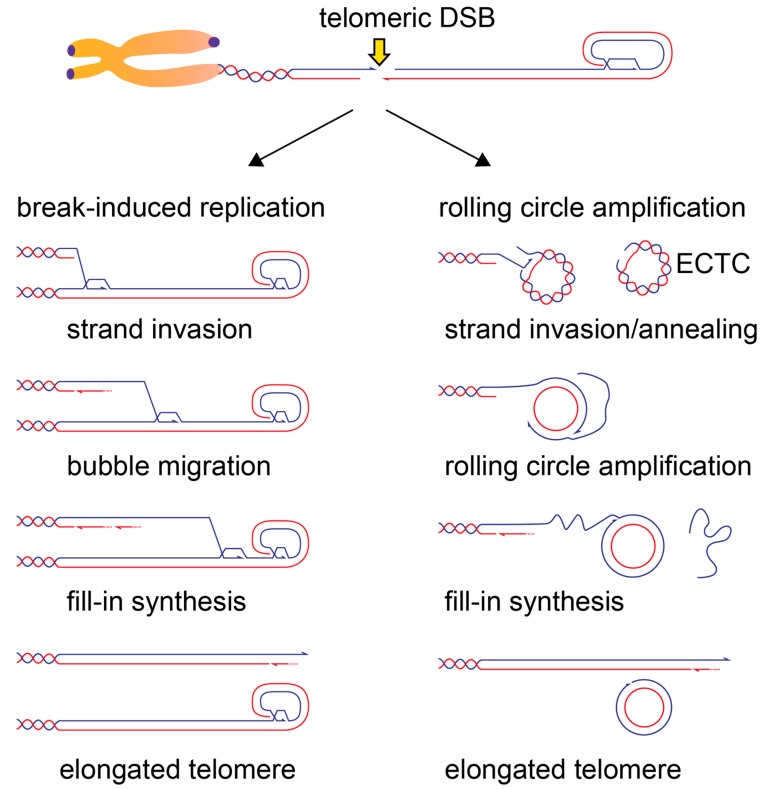
Possible mechanisms of DSB-induced telomere elongation in ALT cells. Telomeric DSB can initiate long-range telomere synthesis. The G-rich strand is represented in purple and the C-rich strand in light blue. Note that for telomere elongation to occur, the 3′ G-rich overhang of the telomere proximal DSB end needs to be engaged in strand invasion. Left: strand invasion of the broken end into a donor telomere will lead to a Break-Induced Replication (BIR)-mediated telomere elongation. A migrating bubble (d-loop) will copy telomeric repeats from the strand invasion point towards the end of the donor telomere. The C-rich strand could then be synthesized through asynchronous lagging strand synthesis, which might involve the use of specialized factors at telomeres, like the CST complex, normally involved in the fill-in of the 3′ overhang [41]. It is not clear what happens when a BIR intermediate encounters the t-loop structure; it is possible that the RTEL1 helicase, which has been implicated in t-loop resolution during replication, plays a similar role in this context. Right: telomeric DSBs could engage in strand invasion/annealing with Extra Chromosomal Telomeric Circles (ECTC) that are abundant in Alternative Lengthening of Telomeres (ALT) cells and often contain single-stranded gaps (C-circles). Strand invasion could proceed in rolling circle amplification that can engage in long-range telomere synthesis while displacing the damaged strand of the C-circle. In both BIR and rolling circle amplification models, telomere elongation occurs in a conservative manner, with both strands being newly synthesized.

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
