# Peer review of "The Response to DNA Damage at Telomeric Repeats and Its Consequences for Telomere Function"

_genes, 2019, doi:10.3390/genes10040318_

Round 1
Reviewer 1 Report
The review is nicely written and will be a good source for catching up with current research in telomere field.
Author Response
Thank you for your kind comments.
Reviewer 2 Report
This is an excellent, timely review of the relationship between telomeres and the DNA damage response pathway. The figures are fantastic and the topics covered are spot-on.
A minor issue is related to a few sentences that need rewarding:
Line 28: redundant wording: a mouse is a rodent.
Line 33: ‘for the simplicity.’
Line 40: ‘initiated’ or ‘activated’ instead of ‘engaged.’
Line 103: This sentence needs rewarding: "Repression of c-NHEJ throughout the telomeric repeats is unlikely to be achieved by de novo formation of a t-loop on the proximal end of a telomeric DSB because c-NHEJ would precede the 5’ end resection required for t-loop formation."
Line 108: instead of ‘seems to involve also the TRF2 interacting factor RAP1’ ‘seems also involve RAP1, the TRF2-interacting factor’.
Line 112 and 114: delete ‘,’
Line 214: "Telomeric replication forks are terminal.. " this sentence need better wording.
Author Response
Reviewer 2
Comments and Suggestions for Authors
This is an excellent, timely review of the relationship between telomeres and the DNA damage response pathway. The figures are fantastic and the topics covered are spot-on.
Thank you for the nice comments and for the suggested corrections.
Here are my answers point by point:
A minor issue is related to a few sentences that need rewarding:
Line 28: redundant wording: a mouse is a rodent.
The sentence was corrected as follows: The POT1 gene has undergone duplication in rodents which express two POT1 proteins, POT1a and POT1b
Line 33: ‘for the simplicity.’
Corrected to: For the sake of simplicity…
Line 40: ‘initiated’ or ‘activated’ instead of ‘engaged.’
“engaged” was changed to “activated”
Line 103: This sentence needs rewarding: "Repression of c-NHEJ throughout the telomeric repeats is unlikely to be achieved by de novo formation of a t-loop on the proximal end of a telomeric DSB because c-NHEJ would precede the 5’ end resection required for t-loop formation."
The original sentence was split in two sentences and modified as follows:
Repression of c-NHEJ throughout the telomeric repeats is unlikely to be achieved by de novo formation of a t-loop on the proximal end of a telomeric DSB. This is because c-NHEJ normally precedes 5’-end resection, which is required to generate the 3’-overhang necessary for t-loop formation.
Line 108: instead of ‘seems to involve also the TRF2 interacting factor RAP1’ ‘seems also involve RAP1, the TRF2-interacting factor’.
Corrected as follows: “seems to involve also RAP1, which is recruited at telomere by TRF2”
Line 112 and 114: delete ‘,’
Deleted.
Line 214: "Telomeric replication forks are terminal.. " this sentence need better wording.
The sentence was changed as follows: Telomeres are replicated by terminal forks (i.e. replication forks moving towards a DNA end)
Reviewer 3 Report
In this mini review, Doksani describes the response to DNA damage at both the inside of telomeres and the chromosome ends, followed by reviewing how various repair pathways, namely c-NHEJ, alt-NHEJ, HR, are involved in repairing telomere damage, and how shelterin proteins may impact these pathways in repairing telomere damage. Finally, the author reviews the pathways that are responsible for telomere maintenance in ALT cells. Overall, this is a very well written and concise mini review with only a few inaccurate claims. The figures are clearly illustrated. Concerns are noted below.
1. Page 1, line 21, remove “for”.
2. Page 1, line 22, “terminate with a 3’ ss overhang 50-400 nt long” should be “terminate with 3’ ss overhangs about 50-400 nt long”.
3. Page 2, line 45, “occurs also” change to “also occurs”.
4. Page 2, line 47, it has been shown that POT1 plays a role in promoting efficient telomere replication (Arnoult et al, 2009, G&D) and POT1 inactivation induces telomere replication stress that leads to activation of ATR (Pinzaru, et al, 2016 Cell Reports). Therefore, it is very likely that ATR activation in POT1-deficient cells is due to replication stress. The author’s statement that such activation is unlikely due to “accumulation from secondary events like replication stress” needs to be modified to include replication stress as a main reason for ATR activation.
5. Page 2, line 56, add “the” in front of “activation of ATM”.
6. Page 6, line 168, “the incidence of T-SCEs results several-fold….” Should be “the incidence of T-SCEs is several fold….”.
7. Page 6, line 202, replace “plays also” with “also plays”.
8. Page 7, line 242 to 244, this sentence is out of context. Why is this sentence there?
9. Page 9, line 290, While both T-circles and C-circles are considered as ECTCs, they are probably two different types of telomere circles that are generated differently. T-circles cannot be detected by the rolling circle amplification assay, and C-circles are not detected by 2D gels. C-circles but not T-circles are detected with rolling circle amplification assay. Please correct.
10. Figure 5, blue and purple lines are difficult to distinguish. Consider using two contrasting colors for two different DNA strands.
Author Response
Reviewer 3
Comments and Suggestions for Authors
In this mini review, Doksani describes the response to DNA damage at both the inside of telomeres and the chromosome ends, followed by reviewing how various repair pathways, namely c-NHEJ, alt-NHEJ, HR, are involved in repairing telomere damage, and how shelterin proteins may impact these pathways in repairing telomere damage. Finally, the author reviews the pathways that are responsible for telomere maintenance in ALT cells. Overall, this is a very well written and concise mini review with only a few inaccurate claims. The figures are clearly illustrated. Concerns are noted below.
I am grateful for the nice comments and the helpful suggestions. Please find my answers point by point below.
1. Page 1, line 21, remove “for”.
Changed as suggested.
2. Page 1, line 22, “terminate with a 3’ ss overhang 50-400 nt long” should be “terminate with 3’ ss overhangs about 50-400 nt long”.
Changed as suggested.
3. Page 2, line 45, “occurs also” change to “also occurs”.
This sentence was modified following the suggestion on the next point.
4. Page 2, line 47, it has been shown that POT1 plays a role in promoting efficient telomere replication (Arnoult et al, 2009, G&D) and POT1 inactivation induces telomere replication stress that leads to activation of ATR (Pinzaru, et al, 2016 Cell Reports). Therefore, it is very likely that ATR activation in POT1-deficient cells is due to replication stress. The author’s statement that such activation is unlikely due to “accumulation from secondary events like replication stress” needs to be modified to include replication stress as a main reason for ATR activation.
I agree, with the reviewer’s comment. It was not my intention to rule out replication stress as a cause of ATR activation in Pot1-deleted cells, but rather to state that Pot1 likely prevents activation of ATR at the telomeric overhang as well. I have included the references suggested from the reviewer that indicate a role of Pot1 in preventing replication stress at telomeres and changed that part as follows:
POT1 has been involved in facilitating telomere replication, therefore ATR activation in POT1-deleted cells could be in part attributable to telomere replication stress [16,17]. However, ATR is activated at telomeres also when POT1 is depleted in G1 cells, indicating that POT1 prevents the exposure of existing single-stranded DNA at telomeres (i.e. the 3’ overhang) [18].
5. Page 2, line 56, add “the” in front of “activation of ATM”.
Changed as suggested
6. Page 6, line 168, “the incidence of T-SCEs results several-fold….” Should be “the incidence of T-SCEs is several fold….”.
Changed as suggested
7. Page 6, line 202, replace “plays also” with “also plays”.
Changed as suggested
8. Page 7, line 242 to 244, this sentence is out of context. Why is this sentence there?
Since sometimes the line numbers do not match perfectly, I assume the reviewer is referring to this sentence:
It is possible that in some cases mutants that increase the rate of stochastic telomere loss are dismissed because of they do not affect the average telomere length.
I have changed it, hoping to make more obvious the connection with the previous sentence. Now it reads as follows:
It is possible that in some cases, mutations in shelterin interactors that increase the rate of stochastic telomere loss, may be dismissed because of a lack of effect on the overall telomere population.
9. Page 9, line 290, While both T-circles and C-circles are considered as ECTCs, they are probably two different types of telomere circles that are generated differently. T-circles cannot be detected by the rolling circle amplification assay, and C-circles are not detected by 2D gels. C-circles but not T-circles are detected with rolling circle amplification assay. Please correct.
While this idea of a clear distinction between t- and c-circles is shared by many in the field, I would like to point out that that it does not have clear experimental basis.
To my knowledge, there is no evidence suggesting that T-circles and C-circles are generated differently.
The only difference between c-circles and t-circles is that c-circles have one or more ssDNA gap that allows them to be amplified efficiently in a rolling circle amplification (RCA) assay in the absence of an added oligonucleotide.
Small gaps would not change substantially the migration properties of C-circles in 2D gels, therefore one cannot distinguish between t- and c- circles in 2D gels. While experiments using native hybridization of 2D-gels indicate that the “t-circle arc” is made of largely double stranded DNA (Nabetani and Ishikawa 2009 PMID: 19015236), they cannot and do not rule out the presence of circles with nicks and or short gaps in that area of the gel.
The initial RCA assays included an oligonucleotide that primes the Phi29 amplification and were therefore designed for the detection of t-circles, as stated by the authors (Zellinger et. al., Mol. Cell 2007 PMID: 17612498) This assay is still widely used in the field (Margalef et. al., Cell 2018 PMID: 29290468).
Finally, there is some confusion on the nomenclature of the ECTC. To my knowledge, the term t-circle was first used in Tomaska et. al., 2004, (PMID 15165907) to refer to telomeric circles in general. The name, t-circle, was adopted also by Tony Cesare and Jack Griffith, that first visualized telomeric circles from ALT cells in electron microscopy and subsequently used by the rest of the field. Roger Reddel’s lab then noticed the existence of C-circles: that are partially single stranded on the G-strand and G-circles (less abundant) that are partially single stranded on the C-strand. Technically, both C-circles and G-circles are t-circles with ss-gaps in one of the two strands.
I have corrected this section of the review, including a sentence that I hope helps to clarify this point. Now it reads as follows:
One of the most consistent features of ALT cells is the presence of Extra-Chromosomal Telomeric Circles (ECTC) that are either double-stranded (commonly referred to as t-circles) or partially single-stranded (commonly referred to as C-circles) [89,134,135]. Despite its common use, t-circle stands for telomeric circles; C-circles are telomeric circles with an intact C-strand and with one or more single-stranded gaps on the G-strand, while G-circles (less abundant) are telomeric circles with an intact G-strand and with one or more single-stranded gaps on the C-strand [135,136].
10. Figure 5, blue and purple lines are difficult to distinguish. Consider using two contrasting colors for two different DNA strands.
I have modified the figures 1 and 5. The lines are now dark blue and red.